# Leucine Repeat Rich Kinase 1 Controls Osteoclast Activity by Managing Lysosomal Trafficking and Secretion

**DOI:** 10.3390/biology12040511

**Published:** 2023-03-29

**Authors:** Sandi Shen, Mingjue Si, Canjun Zeng, Elaine K. Liu, Yian Chen, Jean Vacher, Haibo Zhao, Subburaman Mohan, Weirong Xing

**Affiliations:** 1Musculoskeletal Disease Center, Jerry L Pettis VA Medical Center, Loma Linda, CA 92357, USA; 2Department of Medicine, Loma Linda University, Loma Linda, CA 92354, USA; 3Institut de Recherches Cliniques de Montreal, Montreal, QC H2W 1R7, Canada; 4Département de Médecine, Université de Montréal, Montréal, QC H2W 1R7, Canada; 5Southern California Institute for Research and Education, Long Beach, CA 90822, USA

**Keywords:** Lrrk1, osteoclast, osteopetrosis, bone formation, bone resorption, lysosome, knockout, bone mineral density, acidification, cathepsin K, OSTM1

## Abstract

**Simple Summary:**

Osteoporosis, an age-related disease, develops in part because the rate of new bone formation does not catch up with the increased bone destruction caused by osteoclasts. A full understanding of how osteoclasts function is essential to identify novel drug targets for osteoporosis treatment. Previously, we demonstrated that mice lacking leucine rich repeat kinase 1 (LRRK1) had high bone mass due to defective resorption caused by dysfunctional osteoclasts. Little is known about how LRRK1 regulates osteoclast activity. Here, we found that LRRK1-deficient osteoclasts had an altered distribution of the acidic vacuoles/lysosomes, unlike the wild-type osteoclast. The lysosomes remained in the cytoplasm away from the extracellular lacunae of the bone. The apparent F-actin ring was large, but the sealing zone was weak. Albeit CTSK and v-ATPase were comparably expressed in LRRK1 deficient osteoclasts, CTSK and v-ATPase, two lysosome-associated proteins, were not stationed on the ruffled border. Our data indicate the LRRK1 positively controls osteoclast activity via regulating lysosomal distribution, acid secretion, and protease exocytosis.

**Abstract:**

We previously demonstrated that mice with targeted deletion of the leucine repeat rich kinase 1 (*Lrrk1*) gene were osteopetrotic due to the failure of osteoclasts to resorb bone. To determine how LRRK1 regulates osteoclast activity, we examined the intracellular and extracellular acidification with an acidotropic probe, acridine orange, in live osteoclasts on bone slices. We examined lysosome distribution in osteoclasts by localization of LAMP-2, cathepsin K, and v-ATPase by immunofluorescent staining with specific antibodies. We found that both vertical and horizontal cross-sectional images of the wild-type (WT) osteoclasts showed orange-staining of the intracellular acidic vacuoles/lysosomes dispersed to the ruffled border. By contrast, the LRRK1 deficient osteoclasts exhibited fluorescent orange staining in the cytoplasm away from the extracellular lacunae because of an altered distribution of the acidic vacuoles/lysosomes. In addition, WT osteoclasts displayed a peripheral distribution of LAMP-2 positive lysosomes with a typical actin ring. The clustered F-actin constitutes a peripheral sealing zone and a ruffled border which was stretched out into a resorption pit. The LAMP-2 positive lysosomes were also distributed to the sealing zone, and the cell was associated with a resorption pit. By contrast, LRRK1-deficient osteoclasts showed diffused F-actin throughout the cytoplasm. The sealing zone was weak and not associated with a resorption pit. LAMP-2 positive lysosomes were also diffuse in the cytoplasm and were not distributed to the ruffled border. Although the LRRK1-deficient osteoclast expressed normal levels of cathepsin K and v-ATPase, the lysosomal-associated cathepsin K and v-ATPase were not accumulated at the ruffled border in *Lrrk1* KO osteoclasts. Our data indicate that LRRK1 controls osteoclast activity by regulating lysosomal distribution, acid secretion, and protease exocytosis.

## 1. Introduction

Osteoporosis develops with age in part because the rate of bone formation cannot catch up with the rate of bone loss. Currently, up to 8 million American women (one in five U.S. women) and 2 million American men have osteoporosis, and another 44 million Americans with low bone density are at risk [1]. The total annual healthcare cost for treating osteoporotic fractures was USD 57 billion in 2018, with an additional increase to over USD 95 billion in 2040 [2]. Thus, early treatment of aging-related osteoporosis and prevention of osteoporotic fractures are needed to improve patients’ quality of life and to reduce the healthcare burden.

The ideal drugs for the treatment of osteoporosis should not only promote bone formation but also have an anti-resorptive function. However, such medicines have not been developed yet. Currently, most available anti-osteoporosis drugs such as bisphosphonates function to inhibit bone turnover. However, treatment with bisphosphonates leads to the suppression of both osteoblast-mediated bone formation and osteoclast-mediated bone resorption [3,4,5]. Long-term treatment with these drugs may cause jaw osteonecrosis, increase the risk for atypical fractures of the femur, and compromise fracture repair [6,7,8,9,10,11]. Treatment of osteoporosis with a cathepsin K inhibitor has been reported to cause off-target effects and increase the risk of stroke [12]. Treatment of osteoporotic women with a monoclonal antibody against a receptor activator of nuclear factor-κB ligand (RANKL) was effective; however, the bone loss rebounds after 2 years’ discontinuation [13]. In contrast with the anti-resorptive drugs are the anabolic drugs such as PTH (1–34) and the PTH-related peptide analog, abaloparatide. While these anabolic drugs can promote bone formation, PTH-related drugs need to be injected daily. Given the current limitation of the anabolic drugs and the adverse effects of the anti-resorptive agents, it is necessary to identify new drug targets and develop novel alternative molecules that suppress osteoclast activity without an effect on osteoclast formation.

Leucine rich repeat kinase 1 (LRRK1) belongs to a serine/threonine protein kinase family. The kinase activity relies on GTP binding [14]. A point mutation in the GTP binding domain in LRRK1 caused LRRK1 inaction and dysfunction in vitro [15,16]. Mice lacking the *Lrrk1* gene were severely osteopetrotic in vertebral and long bones due to the functional failure of osteoclasts to resorb bone [17]. Osteoclasts with LRRK1 deficiency were defective in RANKL-induced cytoskeletal reorganization, had weak peripheral sealing zones on bone, and were unable to destroy the bone matrix [17]. *Lrrk1* KO mice responded normally to anabolic PTH treatment and resisted ovariectomy (OVX)-induced bone loss [17]. More recently, an autosomal recessive mutation of *Lrrk1* has been identified in a human patient [17]. A partial deletion of 7th WD-40 repeats and additional amino acid sequence to the C-terminus of LRRK1 protein caused loss of LRRK1 function in osteoclasts [18]. The clinical features of the patient recapitulated the skeletal phenotypes of the *Lrrk1* KO mice. The patient with a loss of LRRK1 function suffered from severe osteopetrosis at the metaphysis of the long and short tubular bones, as observed in the *Lrrk1* KO mice [17,18]. The magnitude of BMD increase in *Lrrk1* KO mice was comparable to osteopetrosis-specific transmembrane protein 1 gene (*Ostm1*) and chloride channel 7 (*Clc7*) KO mouse lines [19,20,21,22]. These studies suggest that LRRK1 may modulate osteoclast function by regulating lysosome distribution, exocytosis, and extracellular acidification in osteoclasts. In this study, we cultured osteoclasts derived from *Lrrk1* KO and WT mice on bone slices and examined lysosome dispersion on the ruffled border and the intracellular and extracellular acidification in osteoclasts in vitro.

## 2. Materials and Methods

### 2.1. Chemicals, Proteins, Plasmids, and Antibodies

Acridine orange (A1301)*,* Alexa fuor-488 conjugated phalloidin (A12379), and protein A/G Sepharose beds were purchased from Invitrogen (Carlsbad, CA, USA). RANKL and macrophage colony-stimulating factor (M-CSF) proteins were purchased from R & D Systems (Minneapolis, MN, USA). Monoclonal and polyclonal anti-actin and anti-flag (M2 and M5) antibodies, polyclonal anti-HA (SAB5600116), monoclonal anti-cathepsin K (MAB3324), and anti-M2 conjugated agarose resin were obtained from Sigma (St. Louis, MO, USA). Monoclonal LAMP2 antibody (ABL-93) was purchased from the Developmental Studies Hybridoma Bank (University of Iowa, Iowa City, IA, USA). Anti-v-ATPase (GTX 32087) antibody was from GeneTex (Irvine, CA, USA). Polyclonal anti-GFP antibody (SC-8334) was from Santa Cruz Biotechnology, Inc. (Santa Cruz, CA, USA). Polyclonal anti-OSTM1 (#14621-1-AP) antibody was from Proteintech (Rosemont, IL, USA). Rhodamine-conjugated second antibodies from mouse, rat, and rabbit (Cat No: 715-025-150, 721-025-150, 711-025-152) were from Jackson ImmunoResearch Laboratory (West Grove, PA, USA). Plasmids of pUC57 harboring mouse Lrrk1, OSTM1-HA, and CLC-7-Myc coding sequences were synthesized by GenScript (Piscataway, NJ, USA) and subcloned into the pRRLsin-cPPT-SFFV-GFP-wpre plasmid (Addgene, Watertown, MA, USA) by replacing GFP. The pRRLsin-cPPT-SFFV-mLrrk1/Flag-wpre was constructed previously [23]. The pcDNA3-OSTM1-eGFP plasmid was generated and kindly provided by our collaborator Jean Vacher at the Laboratory of Cellular Interactions and Development, Clinical Research Institute of Montreal, Montreal, Quebec, Canada, as described [24].

### 2.2. Generation of Lentivirus, Transduction and Transfection

Lentiviral particles were generated by co-transfection of pRRLsin-cPPT-SFFV-OSTM1-wpre plasmid or pRRLsin-cPPT-SFFV-CLC-7-wpre plasmid with pMD2.G and Pax2 plasmids in 293T cells (CRL-3216 ™, American Type Culture Collection) by using FuGene 6 (Promega, Madison, WI, USA) as described by the manufacturer’s instructions. Transduction of primary osteoclast precursors with Lenti-OSTM1 and Lenti-CLC-7 viruses at a ratio of 1:1 was carried out as described previously [16]. For transfection, 2 million trypsinized 293T cells were resuspended in cell line T nucleofector solution (Lonza Bioscience, Hayward, CA, USA) containing 15 µg of pRRLsin-cPPT-SFFV-mLrrk1/Flag-wpre and 7.5 µg of pcDNA3-OSTM1-eGFP plasmids in 2 mm cuvette and electroporated at 110 V for 25 µ-seconds with Gene Pulser Xcell (BioRad, CA, USA). The cells were cultured in 100 mm dishes at 37 °C in a CO_2_ incubator for 24 h, followed by harvesting and lysing in 0.6% CHAPS in 20 mM Bicine buffer (pH 7.5) for immunoprecipitation and Western blot.

### 2.3. Mice, Bone Slice, and Primary Osteoclast Culture

*Lrrk1* KO mice were generated as described [16]. Mice were housed at the VA Loma Linda Healthcare System (VALLHCS) under standard approved laboratory conditions. Animal experiments were performed with the approval of the Institutional Animal Care and Use Committee of VALLHCS (Xing 0005/1442). Tissue-free, midshaft cortical bone of the bovine femur was cut into 100 µm thick slices with an ISOMET LOW Speed Saw (Buehler, IL, USA). The slices were sterilized in 70% ethanol overnight, washed 3 times with PBS, and air dried. The bone slices were exposed to UV light for 20 min and placed in a 48 well-plate with 500 µL α-MEM media over night before use.

Primary monocytes were isolated from the spleen of 4-week-old WT and KO mice as described [17]. The precursors were cultured on bone slices in a 48-well plate in α-MEM supplemented with 10% fetal bovine serum, penicillin (100 units/mL), streptomycin (100 µg/mL), and M-CSF (20 ng/mL) at 37 °C in 5% CO_2_ for 3 days. The cells were infected with lentivirus (Multiplicity Of Infection: 5) and differentiated in the presence of M-CSF (20 ng/mL) and RANKL (50 ng/mL) for 7 days. The media were changed every other day. Multinucleated osteoclasts (e.g., >3 nuclei) were harvested for immunofluorescent staining, cell imaging, or Western blot. TRAP staining and pit formation assays were reported previously [16].

### 2.4. Immunofluorescent Staining of Osteoclasts on Bone Slices

Mature osteoclasts were fixed with 5% formalin in PBS at room temperature for 10 min, then rinsed 3 times with PBS (pH 7.4). The cells were permeabilized with 0.1% saponin, 1% BSA in PBS at room temperature for 30 min. The sections were then incubated with primary antibody at a dilution of 1:300. Negative control cells were incubated with normal rat, rabbit, or mouse IgG. After 1 h incubation at room temperature, the cells were rinsed with PBS and incubated with corresponding rhodamine conjugated secondary antibody at a dilution of 1:200 in PBS containing 5% FBS for an additional 30 min. The cells were washed with PBS again and mounted with Vectashield mounting medium with DAPI (Vector Laboratories, Inc., Burlingame, CA, USA). Actin ring and 3D structure of the osteoclast were scanned and imaged by a confocal laser scanning microscope (Fluoview FV3000, Olympus Corporation, Tokyo 163-0914, Japan).

### 2.5. Immunoprecipitation and Western Blot

293T cells overexpressing mLRRk1/Flag and OSTM1-eGFP were lysed with 0.6% CHAPS in 20 mM Bicine lysis buffer (pH 7.5). The lysate was precleaned with protein A/G Sepharose beds for 1 h at 4 °C, followed by incubation with anti-Flag M2 antibody or control mouse IgG for 1 h. Immunoprecipitation was carried out with protein A/G Sepharose beds at 4 °C overnight. After 4 times washing with lysis buffer, interaction of LRRK1/Flag with OSTM1-eGFP was detected by Western blot using polyclonal anti-GFP antibody. For Western immunoblotting analyses, 20 µg of total cellular lysates were separate on 4–12% SDS-NuPAGE and the protein was transferred to 0.45 µm PVDF membrane. The membrane was incubated with primary antibodies for Western blot as reported [25].

### 2.6. Mass Spectrometer Analyses

Lrrk1 KO and WT osteoclasts were lysed in a lysis buffer containing 50 mM Tris-HCl (pH 8.0), 10 mM MgCl_2_, 150 mM NaCl, 4M guanidine hydrochloride, and 5 mM TCEP-HCl for 20 min at room temperature. Lysate was centrifuged at 12,000 rpm for 10 min and the supernatant was collected. The cellular protein was reduced with 5 mM DTT, and the protein was precipitated with 4 volumes of acetone, followed by overnight trypsin digestion (0.1 mg/mL in 50 mM NH_4_HCO_3_) at 37 °C. Digested peptides were desalted with a Sep-Pac C18 cartridge and the phospho-peptides were enriched with the High- Select ^TM^ TiO_2_ Phospho-peptide Enrichment Kit (Prod#88811, Thermo-Scientific, Rockford, IL 61101, USA) according to the manufacturer’s instruction. The enriched phospho-peptides were dried, resuspended in 0.1% formic acid, and subjected to LS/MS [26].

### 2.7. Acridine Orange Staining

Mature osteoclasts on bone slices were incubated in α-MEM differentiation media containing 5 µg/mL of acridine orange at 37 °C in 5% CO_2_ for 15 min, washed 3 times with the α-MEM medium, and chased in fresh acridine orange-free medium for 10 min, followed by observation under a confocal microscope.

### 2.8. RNA Extraction and Real-Time PCR

RNA was isolated from differentiated osteoclasts [27]. Reverse-transcription was carried out with 300 ng RNA and oligo(dT)_12–18_ in a 20 µL reaction volume to generate cDNA. The real time PCR reaction contained 0.5 µL of cDNA, 1x SYBR GREEN master mix (ABI), and 100 nM of forward and reverse primers in a 12 µL reaction volume. Data were normalized by the expression levels of peptidyl prolyl isomerase A. The primer sequences specific to the mouse *Ostm1* gene are 5′-ACGCAAACTCATTCTACCCAA (forward) and 5′-AGACCTCAGCCTTTTCTGCT (reverse).

### 2.9. Statistical Analyses

Data are expressed as mean ± SEM (standard error of the mean) and were analyzed using Student’s *t*-test. Values were considered statistically significant when *p* < 0.05.

## 3. Results

### 3.1. Lack of LRRK1 in Osteoclasts Distrupts Cytoskeleton Arrangement and Impairs Lysosomal Distribution and Extracellular Acid Secretion

To study the mechanisms of LRRK1 action in bone cells, we labeled the F-actin of active cells on bone with Alexa 488-conjugated phalloidin and analyzed the 3-D structure of osteoclasts under a confocal microscope. We observed that an active WT osteoclast had a typical actin ring, and the cell was associated with a resorptive pit (Figure 1A). By contrast, osteoclasts derived from *Lrrk1* KO mice were large and flat on the bone surface with a disarranged cytoskeleton. The cells were not associated with a resorptive pit (Figure 1B,C). To examine whether proper intracellular and extracellular acidification occurred from the lysosomes in LRRK1 KO osteoclasts, we labeled live osteoclasts with an acidotropic fluorescent probe, acridine orange [28]. The probe emits green fluorescence when bound to dsDNA and orange fluorescence when bound to protons. As shown in Figure 1D,E, we observed significant orange staining of lysosomes in both LRRK1 deficient and WT osteoclasts, suggesting the presence of proper proton production and intracellular acidification. Both vertical and horizontal cross-sectional images of the WT osteoclasts manifested orange staining of the intracellular acidic vacuoles/lysosomes dispersed to the ruffled border (Figure 1D). By contrast, the LRRK1deficient osteoclasts showed fluorescent orange staining in the cytoplasm away from the extracellular lacunae resulted from a change in the distribution of the acidic vacuoles/lysosomes (Figure 1E).

To examine if the lack of LRRK1 in osteoclasts impairs lysosome distribution to the ruffled border, precursors derived from WT and *Lrrk1* KO mice were differentiated on bone slices. Mature osteoclasts were probed with lysosomal-associated membrane protein (LAMP)-2 primary antibody, followed by immunostaining with rhodamine-conjugated second antibody. F-actin of the osteoclasts was stained with Alexa fluor-488 conjugated phalloidin. As shown in Figure 2, WT osteoclast exhibited a peripheral distribution of LAMP-2 positive lysosomes. Cross sections of osteoclasts from both views revealed that clustered F-actin constituted a peripheral sealing zone and a ruffled border that was stretched out into a resorption pit. The LAMP-2 positive lysosomes were also dispersed to the ruffled border, and the cell was associated with a resorption pit. By contrast, osteoclasts defective in LRRK1 showed an apparent F-actin ring with diffused F-actin in the cytoplasm. LAMP-2 positive lysosomes were localized around nuclei and were not fully accumulated at the ruffled border.

It is now well established that protons produced by v-ATPase play a crucial role in lysosomal trafficking and protease exocytosis in osteoclasts. To determine if v-ATPase dysregulation of lysosome distribution affects exocytosis of lysosomal proteases, we determined the cellular distribution of lysosomal cathepsin K (CTSK) and v-ATPase in WT and LRRK1 KO osteoclasts (Figure 3A). We found that in WT osteoclasts, CTSK and v-ATPase were positioned in the ruffled border membrane. Consistent with the impaired lysosomal distribution, however, the lysosomal-associated CTSK and v-ATPase were not positioned in the ruffled border in the LRRK1 KO osteoclasts. The altered distribution of CTSK in the LRRK1 KO osteoclasts cannot be explained on the basis of differences in protein levels since protein levels of CTSK, v-ATPase, and LAMP2, as determined by Western immunoblots, were comparable between LRRK1 KO and WT osteoclasts (Figure 3B, Appendix A). The phenotypes of LRRK1 deficient osteoclasts were very similar to the dysfunctional osteoclasts derived from *Ostm1* and *Clc-7* KO mice [19,20,21,22].

### 3.2. LRRK1 Interacts and Stabolizes Osteoperosis Specific Transmembrane Protein 1 in Osteoclasts

Since LRRK1 belongs to a family of threonine/serine kinases, we theorized that a lack of LRRK1 would impair protein phosphorylation in osteoclasts. To identify the molecular cause for impaired lysosomal distribution in *Lrrk1* KO osteoclasts, we searched for proteins that are differentially phosphorylated between the WT and LRRK1-deficient osteoclasts. Total cellular proteins were extracted from osteoclasts, and the trypsin-digested peptides were then examined for phosphorylated serine/threonine residues by LC/MS analyses. One of the proteins identified was OSTM1, which was phosphorylated at threonine 328 and serine 329 in WT but not LRRK1-deficient osteoclasts (Figure 4A). To study the interaction of LRRK1 with OSTM1, we co-expressed LRRK1-Flag and OSTM1-eGFP in 293T cells by transfection and immunoprecipitated the protein complex with monoclonal anti-Flag M2 antibody. The immunoprecipitated complex was immunoblotted with polyclonal anti-GFP antibody. As shown in Figure 4B & Appendix A, we detected two complexes of LRRK1 with OSTM1-eGFP precursor (75 kD with GFP) and mature OSTM1-eGFP (60 kD with GFP), respectively. These two forms of OSTM1-GFP fusion presented two glycosylated OSTM1 proteins in osteoclasts. We also co-expressed OSTM1-HA and CLC7-Myc in WT and *Lrrk1* KO osteoclasts by lentivirus-mediated transduction and separated the denatured proteins on Phos-tag SDS-PAGE (Wako Chemicals USA, Inc-Fisher). The Phos-tag is a functional molecule that can bind specifically to phosphate groups, which causes retarded migration of the phosphorylated protein on the acrylamide gel [28]. As shown in Figure 4C and Appendix A, retarded phosphorylated OSTM1-HA was detected in WT osteoclasts but not in LRRK1 deficient cells. In addition, total OSTM1 protein appeared reduced, and a small molecular weight peptide was detected in *Lrrk1* KO osteoclasts by Western blot. These studies suggest that activation of LRRK1 signaling might phosphorylate OSTM1 and stabilize OSTM1-HA to prevent it from degradation.

To determine the protein level of endogenous OSTM1 in osteoclasts, total cellular proteins were extracted from WT and LRRK1 deficient osteoclasts for Western blot. Differentiated osteoclasts were also harvested for RNA extraction for real-time RT-PCR. The specificity of the antibody against OSTM1 (Proteintech, #14621-1-AP) was validated prior to use by recognizing the same recombinant OSTM1-HA fusion protein expressed in 293T cells with both anti-HA antibody anti-OSTM1 antibody. With the validated OSTM1 antibody, our immunoblotting analyses showed that the endogenous level of OSTM1 protein was significantly reduced by 80% in LRRK1-deficient mature osteoclasts as compared to the level in WT osteoclasts (Figure 5A,B, Appendix A). However, the transcript of OSTM1 in LRRK1-deficient osteoclasts was increased 1.8-fold compared to the WT osteoclasts (Figure 5C), which could represent a compensatory mechanism for the loss of OSTM1 function in LRRK1-deficient osteoclasts.

### 3.3. Exdogenous OSTM1 Overexpressed in LRRK1 Deficient Osteoclasts Is Not Positioned at the Peripheral Ruffled Border

To determine where OSTM1 protein is distributed in mature osteoclasts, we overexpressed HA-tagged OSTM1 and c-Myc tagged CLC7 in osteoclast precursors by lentivirus infection and differentiated the transduced cells into multinuclear osteoclasts in bone slices. The OSTM1 expression and osteoclast distribution were examined by immunofluorescent staining with specific antibodies against HA, followed by rhodamine-conjugated second antibody staining. F-actin of the osteoclast was stained with Alexa flor-488 conjugated phalloidin. As shown in Figure 6A, WT osteoclasts were able to transport OSTM1 protein-containing lysosomes to the peripheral ruffled border and were associated with a resorptive pit. Consistent with the altered lysosomal distribution in LRRK1 deficient cells, a low level of OSTM1 protein is dispersed in the cytoplasmic compartment of the LRRK1 deficient osteoclast. The *Lrrk1* KO osteoclasts were unable to accumulate OSTM1-containing lysosomes on the ruffled border. In addition, the LRRK1-deficient osteoclasts were not associated with a resorptive pit on bone slices with enlarged vacuoles in the cytoplasmic compartment (Figure 6A). Our TRAP staining and the pit assay confirmed that overexpression of OSTM1/CLC7 in LRRK1-deficient osteoclasts were TRAP-positive multinuclear cells, but they were dysfunctional. Unlike the WT osteoclasts, these cells resorbed little bone in a pit formation assay (Figure 6B). These data strongly suggest that OSTM1 phosphorylation initiated by LRRK1–OSTM1 interaction may be essential for osteoclast function that stabilizes the OSTM1/CLC7 complex and facilitates lysosome trafficking and acid secretion.

## 4. Discussion

Of over 4500 gene KO mouse lines, *Lrrk1*-ablated mice exhibited the highest bone mass [29]. Mice with deletion of *Lrrk1* manifested an osteopetrosis phenotype [17]. Little is known about the mechanism of LRRK1 action in osteoclasts. Recent studies reported that LRRK1 could modulate EGFR trafficking and lysosomal degradation via phosphorylating CLIP-170 and promoting CLIP-170-mediated cytoskeleton rearrangement in HEK293 cells [30,31]. However, a change in EGFR activation is known to influence osteoclast formation and survival [32]. Mice with disruption of EGFR had low bone mass due to a decreased bone formation and an increase in bone resorption [33], which were the opposite of what was observed in the *Lrrk1* KO mice. More recently, LRRK1 has been found to phosphorylate DCK5-RAP2 to promote CDK5-RAP2 interaction with γ-tubulin [34]. However, mice with dysfunctional CDK5-RAP2 were small, had severe anemia, and died after birth. The phenotypes were inconsistent with the skeletal manifestations of *Lrrk1* KO mice [17,35,36]. Thus, it is unlikely that CLIP-170 and CDK5-RAP2 are involved in LRRK1 regulation of osteoclast function in bone. In this study, we performed proteomics analyses to determine how LRRK1 regulates osteoclast activity and examined the cytoskeleton organization of active osteoclasts derived from *Lrrk1* KO and WT control mice. We found that LRRK1 controls osteoclast activity by modifying OSTM1 function to regulate lysosomal distribution in osteoclasts.

The ability of osteoclasts to resorb bone largely relies on cytoskeletal rearrangement and exocytosis in the ruffled border. The integrin αvβ3 and downstream signal molecules of c-Src, Syk, Slp-76, Vav3, and Rac are known to regulate the cytoskeleton rearrangement [37,38,39,40,41]. Disruption of any of these molecules in osteoclasts compromises cytoskeleton remolding and bone resorption. Our previous studies demonstrated that Src, RAC1/2/Cdc42 proteins, and L-plastin were involved in regulation of LRRK1 function in osteoclasts [17,26,42]. However, mice with ablation of these genes displayed milder bone phenotypes than the *Lrrk1* KO mice, suggesting the possibility of alternate substrates of LRRK1 in osteoclasts besides these signaling molecules [38,40,43]. Recently, it was reported that LRRK1 phosphorylated RAB7 at serine 72 residue in embryo fibroblasts and NDEL1 at serine 155 in epithelial cells, respectively [44,45]. LRRK1 phosphorylation of RAB7 led to recruitment of the dynein–dynactin complex to RAB7-positive vesicles, and as such, facilitated the dynein-driven transport of the EGFR-containing endosomes toward the perinuclear region [46]. Lack of RAB7 expression in cultured osteoclasts disrupted the polarization of the osteoclasts and the targeting of vesicles to the ruffled border [47]. However, the exact role of RAB7 in the regulation of osteoclast function in vivo is unknown since KO of *Rab7* gene in mice was embryonically lethal [45]. Missense mutations in *Rab7* in humans have been associated with Charcot-Marie-Tooth type 2B neuropathy [48]. Interestingly, a protein complex formation containing RAB7, EDF8, FAM98A, NDEL1, and PLEKHM1 in osteoclasts has been reported to be critical for peripheral lysosome distribution and bone resorption [49]. It remains to be determined whether LRRK1 phosphorylation of RAB7 and NDEL1 also happens in osteoclasts to facilitate lysosome trafficking and stabilize the protein complex. However, based on the much milder bone phenotypes of osteoclast specific conditional *Plekhm1* and *Ndel1* KO mice compared with *Lrrk1* KO mice, it is unlikely that PLEKHM1/RAB7/NEDL1 complex plays a major role in the regulation of LRRK1 function in osteoclasts and bone resorption in mice [49,50]. 

Bone resorption relies on the extracellular acidification of vacuolar ATPase proton pumps positioned in the ruffled border of osteoclasts. In this study, we examined the lysosome distribution, the CTSK and v-ATPase localizations, and the intracellular and extracellular acidification in live osteoclasts. We found that both WT and KO osteoclasts have normal intracellular acid production, as evidenced by the presence of orange staining vacuoles in the cytoplasmic compartment. The WT osteoclast displayed orange staining of the intracellular acidic vacuoles/lysosomes positioned to the ruffled border toward the extracellular acidification. By contrast, the osteoclast deficient LRRK1 manifested fluorescent orange staining in the cytoplasm away from the extracellular lacunae, indicating that the distribution of the acidic vacuoles/lysosomes were altered in the absence of LRRK1. In addition, the WT osteoclasts demonstrated a peripheral distribution of LAMP-2 positive lysosomes with a typical actin ring. The clustered F-actin formed a peripheral sealing zone and a ruffled border that was expanded into a dent resorption pit. The LAMP-2 positive lysosomes were also distributed to the sealing zone and dug into the resorbed pit. By contrast, osteoclasts lacking LRRK1 showed diffused F-actin in the cytoplasm. The sealing zone was weak, and the cell membrane was not extruded into the bone. LAMP-2 positive lysosomes were also diffuse in the cytoplasm and were not distributed to the ruffled border. While no change in the expression levels of CTSK and V-ATPase was found in the LRRK1 deficient cells compared to those in WT osteoclasts, CTSK and v-ATPase, which are associated with lysosomes, were not positioned in the ruffled border in *Lrrk1* KO osteoclast. Our studies indicate that LRRK1 signaling positively regulates the lysosomal position on the ruffled border, acid secretion, and protease exocytosis in osteoclasts. Our findings in bone cells are inconsistent with a recent report that LRRK1 negatively regulates vesicle-associated membrane protein 7 (VAMP7)-mediated lysosomal exocytosis in COS-7 cells [51], suggesting that LRRK1 function is cell-type dependent. If LRRK1 negatively regulates VAMP7-mediated lysosomal exocytosis in osteoclasts as reported in COS-7 cells, we would then predict that lack of LRRK1 would cause more exocytosis and increased bone resorption. By contrast, our studies found that deficiency of LRRK1 in osteoclasts resulted in dysfunction of exocytosis and reduced bone resorption. While our data support the impaired exocytosis in LRRK1 deficient osteoclasts, further studies, including measurements of the intracellular Cl^−^ concentration, and pH in osteoclasts placed on bone slices and the secreted CTSK levels in the culture media derived from LRRK1 deficient osteoclasts are needed to confirm our findings.

Our proteomics study identified that OSTM 1, a β-subunit of the chloride channel 7 (ClC7), was highly phosphorylated in WT osteoclasts but not in Lrrk1 KO cells. We also demonstrated that LRRK1 protein physically interacted with OSTM1, and the lack of LRRK1 protein in osteoclasts caused a reduction in OSTM1 level and disposition of OSTM1 to the ruffled border. It is known that phosphorylation can modulate the nature and the strength of protein–protein interactions by either affecting the binding energy of the complex or changing the protein conformation [52]. Our data are consistent with previously described lysosomal distribution in LRRK1-deficient cells and suggest that OSTM1 is a downstream factor of the LRRK1 signaling pathway in osteoclasts. Phosphorylation of threonine/serine at the 328/329 residues of OSTM1 in LRRK1 expressing osteoclasts may stabilize OSTM1 and lysosomal proton secretion into the extracellular lacunae, thus providing an acidic microenvironment required for bone resorption. In support of the key involvement of OSTM1 in LRRK1 signal pathways are the findings that both OSTM1 and CLC7 play a critical role in regulating osteoclast function by providing extracellular acidification and ablation of either *Ostm1* or *Clc7* in mice, causing severe osteopetrosis [19]. The C-terminal region of OSTM1 has been known to interact with kinesin 5B, a motor for centrifugal distribution of lysosomes in osteoclasts. The lack of OSTM1 vesicles linked to KIF5B cargos caused an absence of lysosomes on the ruffled border, resulting in dysfunctional secretion, inefficient bone resorption, and osteopetrosis [24]. More interestingly, the magnitude of BMD increase in *Lrrk1* KO mice was comparable to *Ostm1* and *Clc7* KO mouse lines [19,20,21,22]. Our data, together with others’ findings that OSTM1-deficient mice and patients with *Ostm1* mutations exhibited severe osteopetrosis due to osteoclast dysfunction [20,21,22,53], strongly indicate that OSTM1 phosphorylation and interaction with LRRK1 may play a critical role in the regulation of osteoclast function via modulating lysosomal peripheral movement and acid secretion on the ruffled border.

In this study, we did not determine whether the OSTM1 is the direct biological substrate of LRRK1 nor examine how LRRK1 stabilizes OSTM1 protein in osteoclasts. While we detected phosphorylated OSTM1 in WT osteoclasts, the lack of detection of phosphorylated OSTM1 protein in the LRRK1 deficient cells could be due to the OSTM1 destabilization and degradation rather than the reduced phosphorylation of OSTM1. Consistent with this interpretation, our Western blot and gene expression studies demonstrated an 80% reduction in the endogenous OSTM1 protein and an approximately 2-fold compensatory increase in the mRNA expression of *Ostm1* in LRRK1-deficient osteoclasts. Therefore, it is likely that LRRK1 stabilization of OSTM1 plays a critical role in regulating osteoclast function either by their physical interaction or by phosphorylation of OSTM1 to enhance OSTM1 interaction with CLC7 or both. The relative contribution of LRRK1/OSTM1 interaction vs. LRRK1 phosphorylation of OSTM1 to OSTM1 stabilization needs to be studied in the future. We will evaluate whether the skeletal phenotypes in the *Lrrk1* KO mice can be rescued by transgenic overexpression of OSTM1 or expression of phosphomimic OSTM1 in osteoclasts. We will also investigate if the OSTM1 is a direct biological substrate of LRRK1 using purified recombinant proteins in an in vitro kinase assay.

## 5. Conclusions

In this study, we found that the LRRK1-deficient osteoclast, unlike the WT osteoclast, has an altered distribution of the acidic vacuoles/lysosomes. The lysosomes remained in the cytoplasm away from the extracellular lacunae of the bone in *Lrrk1* KO osteoclasts. The apparent F-actin ring was large, but the sealing zone was weak. Although the LRRK1-deficient cells expressed comparable CTSK and v-ATPase protein levels, CTSK and v-ATPase were not stationed on the ruffled border in LRRK1-deficient osteoclasts. Our data indicate the LRRK1 positively modulates osteoclast activity via controlling lysosomal distribution, acid secretion, and exocytosis. Further studies are needed to evaluate if the skeletal phenotypes in the *Lrrk1* KO mice can be rescued by transgenic overexpression of OSTM1 or expression of phosphomimic OSTM1 in osteoclasts.

## Figures and Tables

**Figure 1 biology-12-00511-f001:**
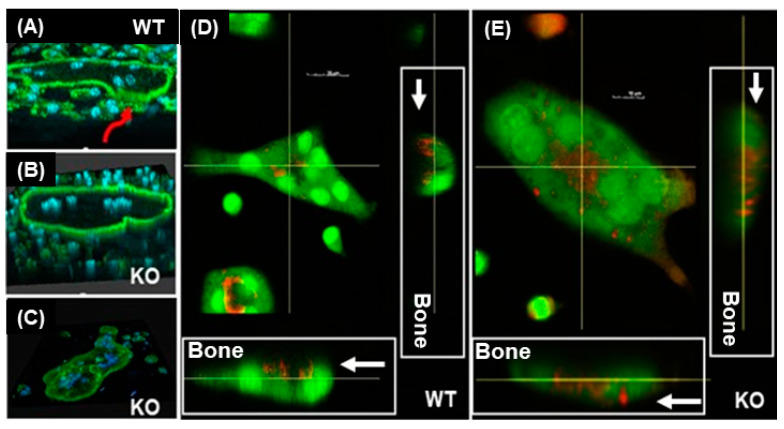
Lack of LRRK1 in osteoclasts disrupts cytoskeletal rearrangement and impairs extracellular acidification. Osteoclasts derived from 4-week-old male mice were cultured on bone slices and labeled with Alexa Fluor-488 conjugated phalloidin and DAPI (**A**–**C**) or acridine orange (**D**,**E**). Staining was analyzed by confocal microscopy. (**A**) A wild-type (WT) osteoclast formed a clear actin ring associated with a pit on a bone slice as the red arrow indicates. (**B**,**C**) 3-D images of LRRK1 knockout (KO) osteoclasts with a disrupted cytoskeleton and an apparent actin ring, respectively, without association with pits. (**D**,**E**) Horizontal and vertical cross-section images of the live WT and LRRK1 KO osteoclasts stained with the vital fluorescent dye acridine orange. Arrows indicate acridine orange-stained protons from the lysosomes (orange).

**Figure 2 biology-12-00511-f002:**
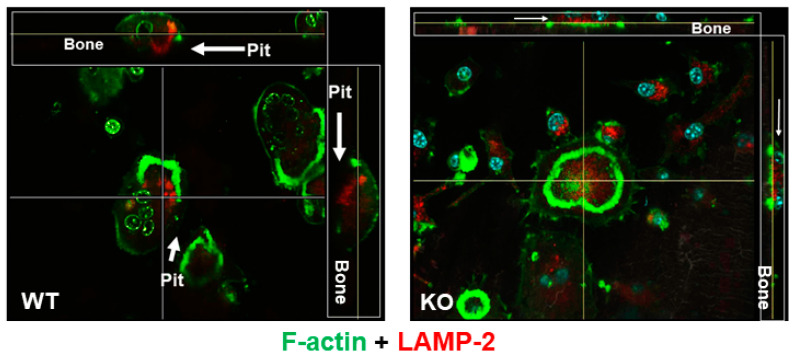
Lack of LRRK1 in osteoclasts impairs lysosome distribution to the ruffled border. Differentiated osteoclasts on bone slices were probed with monoclonal anti-lysosomal associated membrane protein 2 (LAMP-2) antibody from rat, followed by staining with rhodamine-conjugated anti-rat IgG second antibody. F-actin of the osteoclasts was stained with Alexa fluor-488-conjugated phalloidin. Immunofluorescent staining was analyzed on a confocal microscope. The two lines in the middle of the cell represent the positions of horizontal and vertical cuts, respectively. Arrows indicate F-actin and LAMP-2 associated lysosomes with a dent on the horizontal and vertical cross-sections of WT and LRRK1-deficient osteoclasts, respectively.

**Figure 3 biology-12-00511-f003:**
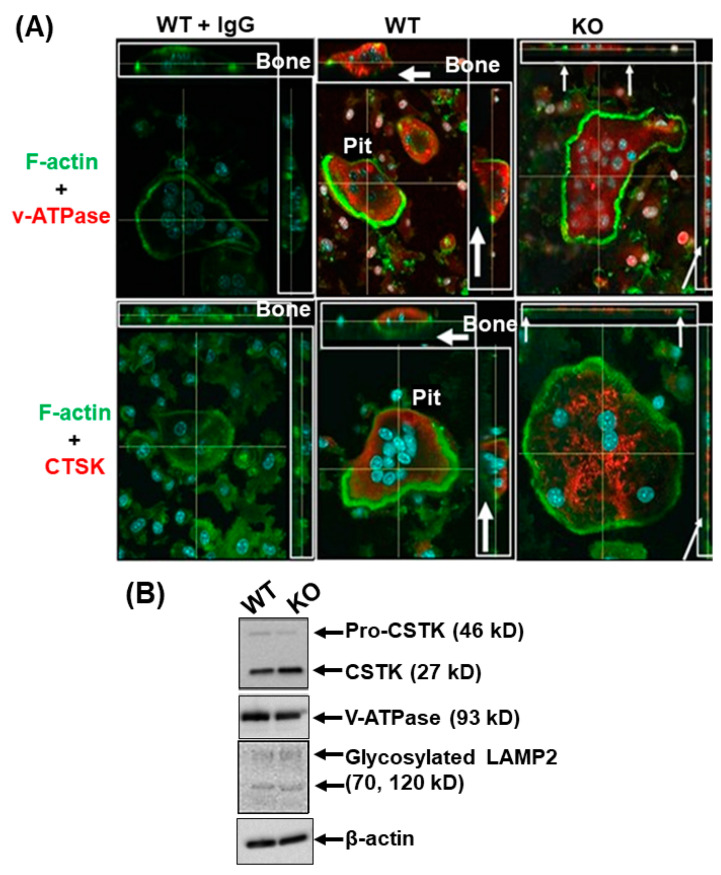
LRRK1-deficient osteoclasts cannot localize cathepsin K (CTSK) and V-ATPase to the ruffled border. (**A**) Differentiated osteoclasts on bone slices were probed with monoclonal anti-CTSK from mouse and polyclonal anti-V-ATPase antibodies from rabbit or corresponding control IgG, respectively, followed by staining with rhodamine-conjugated second antibody. F-actin of the osteoclasts was stained with Alexa fluor-488-conjugated phalloidin. Nuclei were stained with DAPI. Immunofluorescent staining was analyzed by a confocal microscope. The two lines in the middle of the cell represent the positions of horizontal and vertical cuts, respectively. Arrows indicate F-actin and CTSK, and V-ATPase, respectively, associated with a pit on the horizontal and vertical cross-sections of WT but not LRRK1-deficient osteoclasts. (**B**) Monoclonal antibodies against CTSK and LAMP-2, and polyclonal antibodies specific to v-ATPase, were validated by Western blot. Comparable levels of CTSK, v-ATPase, and LAMP-2 were expressed in LRRK1-deficient and WT control osteoclasts.

**Figure 4 biology-12-00511-f004:**
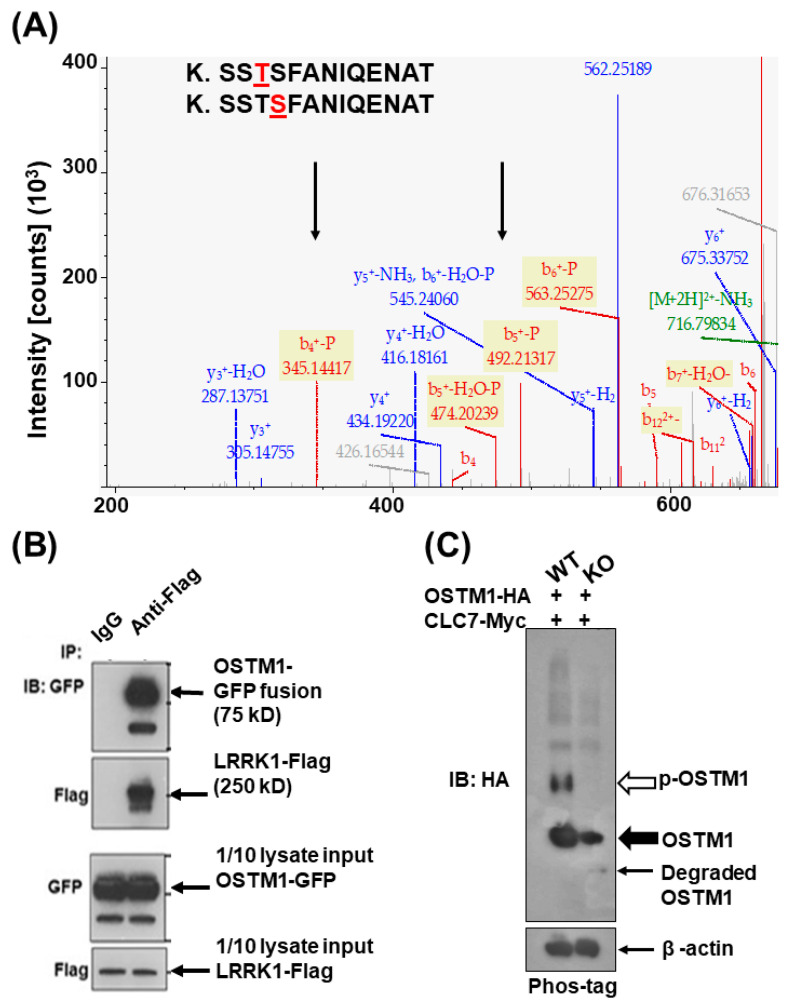
LRRK1 interacts with and stabilizes OSTM1 in osteoclasts. (**A**) Phosphorylation of OSTM1 at threonine 328 and serine 329 was detected in WT osteoclasts but not in LRRK1-deficient osteoclasts by LC/MS. (**B**) LRRK1 interacts with OSTM1-GFP. Complex formation of LRRK1–Flag with OSTM1–GFP was detected by immunoprecipitation (IP) with an anti-Flag antibody, followed by immunoblotting (IB) with an anti-GFP antibody. (**C**) Phosphorylated OSTM1 is present in WT osteoclasts. Phosphorylated proteins were resolved on 7% Phos-tag SDS-PAGE. Filled and open arrows indicate unmodified OSTM1-HA and phosphorylated OSTM1-HA, respectively. A small molecular weight peptide of degraded OSTM1 was detected only in LRRK1-deficient cells, as indicated by an arrow. The protein marker was not run on a Phos-tag PAGE.

**Figure 5 biology-12-00511-f005:**
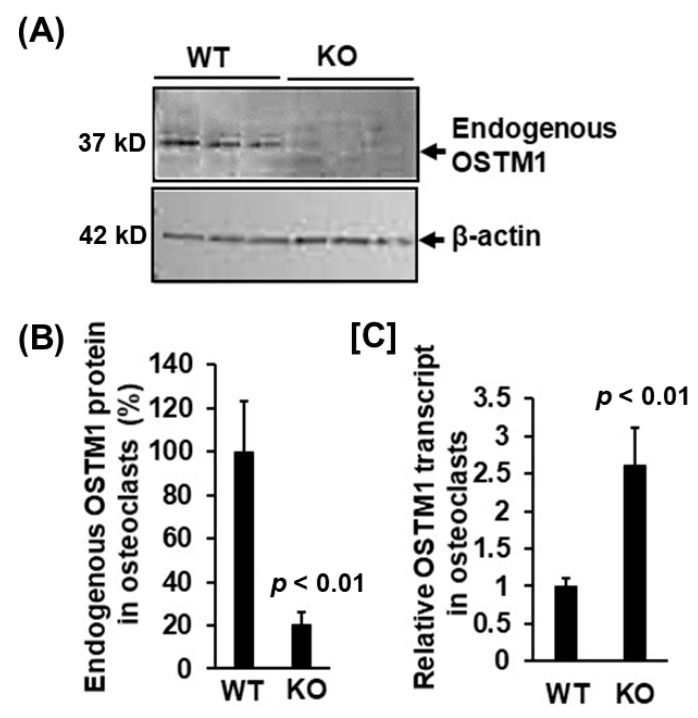
Endogenous OSTM1 protein is reduced in LRRK1-deficient osteoclasts. (**A**) Western blot with an anti-OSTM1 antibody detected a reduced level of endogenous OSTM1 in LRRK1 deficient osteoclasts. (**B**) Quantitative data of Western blot (N = 6, 4 weeks old males and females). (**C**) Increased transcript of OSTM1 in osteoclasts, measured by Real-time RT-PCR (N = 4, males and females).

**Figure 6 biology-12-00511-f006:**
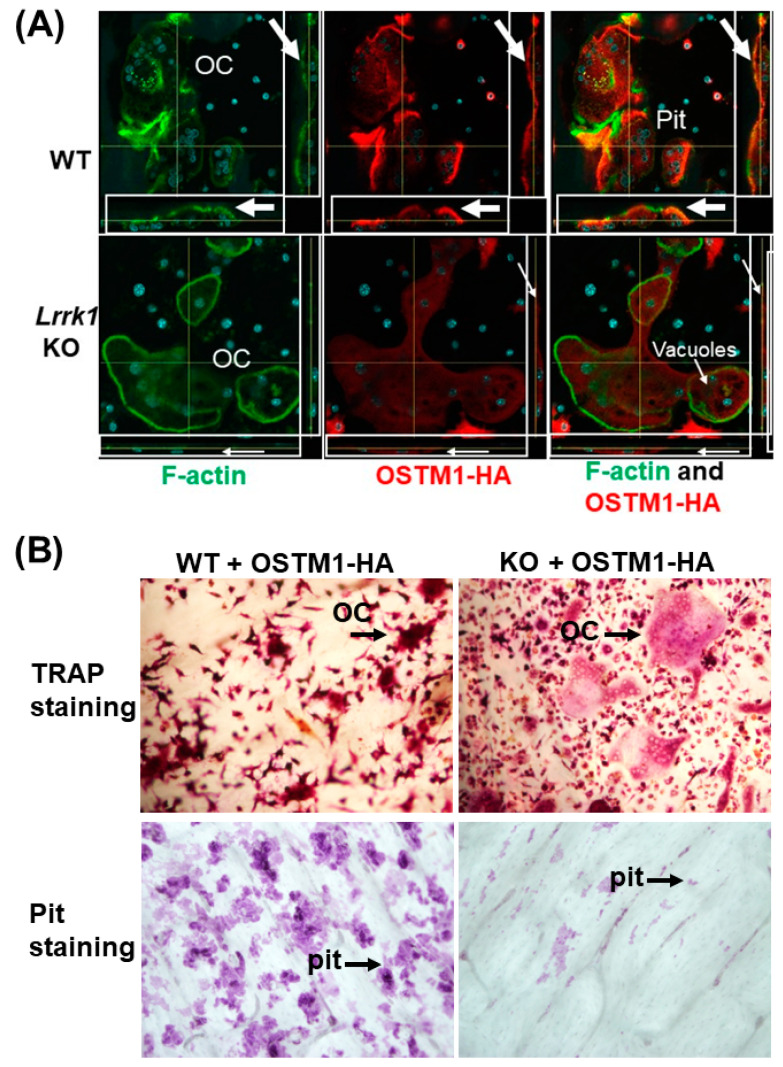
LRRK1-deficient osteoclasts are not able to transport OSTM1 to the peripheral ruffled border. (**A**) Differentiated osteoclasts on bone slices were probed with anti-HA antibody, followed by staining with a rhodamine-conjugated second antibody from rabbit. F-actin of the osteoclasts was stained with Alexa fluor-488 conjugated phalloidin. Nuclei were stained with DAPI. Immunofluorescent staining was analyzed by a confocal microscope. The two lines in the middle of the cell represent the positions of horizontal and vertical cuts, respectively. Arrows indicate F-actin and HA-OSTM1 fusion protein associated with a pit on the horizontal and vertical cross-sections of WT but not LRRK1-deficient osteoclasts. Enlarged vacuoles were frequently seen in LRRK1 deficient osteoclasts. (**B**) Images of active WT and LRRK1 deficient osteoclasts (OCs) stained with TRAP and the resorbed bones stained with hematoxylin for pits (100×). The osteoclasts derived from the WT and *Lrrk1* KO mice were transduced with Lenti-OSTM1-HA and Lenti-CLC7-myc viruses.

## Data Availability

Data presented in this study are availability from PI upon request.

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
