# Peer review of "Leucine Repeat Rich Kinase 1 Controls Osteoclast Activity by Managing Lysosomal Trafficking and Secretion"

_biology, 2023, doi:10.3390/biology12040511_

Round 1

Reviewer 1 Report

In this study, Shen et al. used splenic cells from LRRK1-knockout and wildtype mice as osteoclast precursors. They turned osteoclast precursor cells seeded on bone slices to osteoclasts in vitro, with the help from M-CSF and RANKL. Through various experiments, the authors confirmed their precious finding that LRRK1-null osteoclasts are dysfunctional. The authors further investigated the causes of the reduced resorptive activity of LRRK1-null osteoclasts. They revealed that osteoclasts lack of LRRK1 with abnormal lysosomal distribution, acid secretion, and protease exocytosis. This is an interesting study. However, their manuscript is not well written.

The authors should avoid using awkward (line 77, 222-224, etc.) or long sentences (line 212-217, etc.). The authors should correct grammar mistakes (such as line 220 an acidotropic probe, not “the”; line 319 and line 326 unnecessary “the”; line 347 on but “in” bone slices; line 365 “were staining with”). The authors should provide the origin of materials such as bone slices used for their study. The authors should cite most recent information (line 61). The authors should describe their experimental methods precisely, not like line 234, which is conflicted with what they described in line 148-149. The authors should ensure their figures more organized and annotated (such as Fig. 2 image identity; Fig. 4B immunoprecipitates or total cell lysates; Fig. 4C what pointed by “filled and open arrows” are not just OSTM1). The authors must rewrite their Discussion section, because line 371-420 is not part of “Discussion”. Line 230 “seemed much larger” is an improper description, the authors must quantify the difference between the two cell types.

Author Response

We thank the reviewer for his/her positive remarks and helpful suggestions.

  • The authors should avoid using awkward (line 77, 222-224, etc.) or long sentences (line 212-217, etc.).

Response: We have revised these sentences, as recommended.

  • The authors should correct grammar mistakes (such as line 220 an acidotropic probe, not “the”; line 319 and line 326 unnecessary “the”; line 347 on but “in” bone slices; line 365 “were staining with”).

Response: We apologize for the typos and errors. We have corrected the errors in the revised manuscript.

  • The authors should provide the origin of materials such as bone slices used for their study.

Response: We apologize for the missing details on materials. We have now added the missing part in the materials and methods. We stated that “Tissue-free cortical bone of the femur from bovine was cut into 100 μm thick slices with an ISOMET LOW Speed Saw (Buehler, IL, USA). The slices were sterilized with 70% ethanol overnight, washed 3 times with PBS, and air dried. The bone slices were exposed to UV light for 20 minutes and placed in a 48 well-plate with 500 µl αMEM media over night before use before use.”

  • The authors should cite most recent information (line 61).

Response: We have updated information in the revised manuscript.

  • The authors should describe their experimental methods precisely, not like line 234, which is conflicted with what they described in line 148-149.

Response: We apologize for the inconsistency. We have now corrected the error in the revised manuscript. We stated that “Monocytes were isolated from 4-week-old mice.”

  • The authors should ensure their figures more organized and annotated (such as Fig. 2 image identity; Fig. 4B immunoprecipitates or total cell lysates; Fig. 4C what pointed by “filled and open arrows” are not just OSTM1). 

Response: We apologize for the missing labels. We have now added the labels in the revised manuscript.

  • The authors must rewrite their Discussion section, because line 371-420 is not part of “Discussion”. Line 230 “seemed much larger” is an improper description,

Response: We have extensively revised the discussion.

  • the authors must quantify the difference between the two cell types.

Response: Differences in the expression levels both mRNA and protein were quantified in the revised manuscript.

Reviewer 2 Report

Several minor revising are needed:

The use of “293T cells” were mentioned in Line 133 and Line 308, authors shall provide information about source of this cell line. We know this cell line has a high transfection efficiency, and authors have achieved good results. Therefore, readers might like to use the same company’s product.

Figure 4 A is not readable, authors may consider increasing the size. 

Author Response

We thank the reviewer for her/his positive remarks. We have added the source of 293T cells in the revised manuscript. We have also revised Figure 4 in the revised manuscript. 

Reviewer 3 Report

This work by Shen et al. explores the role of LRRK1 in osteoclastic activity. The authors report that deletion of Lrrk1 gene in primary spleen monocytes-derived osteoclasts changes the distribution of the acidic vacuoles/lysosomes which remain in the cytoplasm and not in the bone extracellular lacunae, unlike wild-type osteoclasts. Although F-actin rings are formed in KO osteoclast and cells express CTSK and V-ATPase like normal cells, these are not localized in the ruffled border which impairs bone resorption. The authors conclude that LRRK1 modulates LAMP-2 positive lysosomal distribution and acid/CTSK secretion resulting in disrupted osteoclastic activity.

The work is very interesting and the manuscript is well-written. The authors used a wide range of experimental approaches to obtain the results and draw conclusions. Some points need to be addressed.

1.       Line 29, replace disruption with deletion.

2.       How was the multiplicity of infection was determined? Was cytotoxicity assessed following lentiviral infection?

3.       In Methods 2.3, it is reported that mice were 5-6 weeks old while in the legends to figures it is written that mice were 4 weeks. This needs clarification.

4.       Why both genders were used? Was any difference observed in osteoclastic activity between males and females?

5.       Fig2. WT and KO are not labeled.

6.       Line 298, stabilizes.

7.       Since the sample number is small, scatter plots with bars should be used for all quantifications.

8.       What type of bone slices were used? TRAP-stained bone slices need to be added, e.g. in Fig6, to show the effect on the actual bone resorption.

Author Response

We thank the reviewer for her/his positive remarks.

  1. Line 29, replace disruption with deletion.

Response: We have replaced the word disruption with deletion in the revised manuscript.

  1. How was the multiplicity of infection was determined? Was cytotoxicity assessed following lentiviral infection?

Response: We apologize for the missing information. We have now added MOI 5 in the revised manuscript. We did not observe cytotoxicity after transduction and the cells were fused to form mature osteoclasts.

  1. In Methods 2.3, it is reported that mice were 5-6 weeks old while in the legends to figures it is written that mice were 4 weeks. This needs clarification.

Response: We apologize for the mistake. We have now corrected the inconsistency in the revised manuscript.

  1. Why both genders were used? Was any difference observed in osteoclastic activity between males and females?

Response: We did evaluate the phenotypes in females and males, respectively based on the NIH experimental guidelines. Since we did not see a significant gender difference between genotypes in vitro assays, we combined the data of males and females for quantification of Western blots.

  1. WT and KO are not labeled.

Response: We apologize for the missing labels. We have now added the labels in the revised manuscript.

  1. Line 298, stabilizes.

Response: We apologize for the error. We have now corrected the error in the revised manuscript.

  1. Since the sample number is small, scatter plots with bars should be used for all quantifications.

Response: We have used the bars with mean ± SEM as stated in the materials and methods.

  1. What type of bone slices were used? TRAP-stained bone slices need to be added, e.g. in Fig6, to show the effect on the actual bone resorption.

Response: We apologize for the missing information. We have now added how the bone slices were prepared in the revised manuscript. We agree with the reviewer that TRAP staining slices need to be added. However, the TRAP staining of osteoclasts only showed the mature multinuclear cells. It does not show the resorption. Thus, we added hematoxylin stained bone slices to show the resorption pits in figure 6 in the revised manuscript.